# Implantation of Various Cell-Free Matrixes Does Not Contribute to the Restoration of Hyaline Cartilage within Full-Thickness Focal Defects

**DOI:** 10.3390/ijms23010292

**Published:** 2021-12-28

**Authors:** Shabnam I. Ibragimova, Ekaterina V. Medvedeva, Irina A. Romanova, Leonid P. Istranov, Elena V. Istranova, Aleksey V. Lychagin, Andrey A. Nedorubov, Peter S. Timashev, Vladimir I. Telpukhov, Andrei S. Chagin

**Affiliations:** 1Institute of Regenerative Medicine, I.M. Sechenov First Moscow State Medical University (Sechenov University), 119991 Moscow, Russia; shabnam.olimp97@gmail.com (S.I.I.); medvedevaekaterina@yandex.ru (E.V.M.); istranov_l_p@staff.sechenov.ru (L.P.I.); istranova_e_v@staff.sechenov.ru (E.V.I.); timashev_p_s@staff.sechenov.ru (P.S.T.); 2World-Class Research Center, Digital Biodesign and Personalized Healthcare, Sechenov University, 8-2 Trubetskaya St., 119991 Moscow, Russia; romirandreevna@gmail.com; 3Department of Traumatology, Orthopedics and Disaster Surgery, Sechenov University, 8-2 Trubetskaya St., 119991 Moscow, Russia; lychagin_a_v@staff.sechenov.ru; 4Center for Preclinical Research, Institute of Translational Medicine and Biotechnology, I.M. Sechenov First Moscow State Medical University, 119991 Moscow, Russia; nedorubov_a_a@staff.sechenov.ru; 5Department of Operative Surgery and Topographic Anatomy, I.M. Sechenov First Moscow State Medical University (Sechenov University), 119991 Moscow, Russia; 6Department of Physiology and Pharmacology, Karolinska Institutet, 17177 Stockholm, Sweden

**Keywords:** articular cartilage, full-thickness defect, scaffold, collagen membrane, decellularized cartilage, cellulose

## Abstract

Articular cartilage is a highly organized tissue that has a limited ability to heal. Tissue engineering is actively exploited for joint tissue reconstruction in numerous cases of articular cartilage degeneration associated with trauma, arthrosis, rheumatoid arthritis, and osteoarthritis. However, the optimal scaffolds for cartilage repair are not yet identified. Here we have directly compared five various scaffolds, namely collagen-I membrane, collagen-II membrane, decellularized cartilage, a cellulose-based implant, and commercially available Chondro-Gide^®^ (Geistlich Pharma AG, Wolhusen, Switzerland) collagen membrane. The scaffolds were implanted in osteochondral full-thickness defects, formed on adult Wistar rats using a hand-held cutter with a diameter of 2.0 mm and a depth of up to the subchondral bone. The congruence of the articular surface was almost fully restored by decellularized cartilage and collagen type II-based scaffold. The most vivid restoration was observed 4 months after the implantation. The formation of hyaline cartilage was not detected in any of the groups. Despite cellular infiltration into scaffolds being observed in each group except cellulose, neither chondrocytes nor chondro-progenitors were detected. We concluded that for restoration of hyaline cartilage, scaffolds have to be combined either with cellular therapy or morphogens promoting chondrogenic differentiation.

## 1. Introduction

Hyaline cartilage covers joint surfaces and provides amortization as well as slide facilitation during joint movements. The matrix of this highly organized tissue consists mostly of type II collagen fibers, proteoglycans, and hyaluronic acid, but can also contain elastin, collagen type IX, and other minor components [1]. The nerve endings, blood vessels, and lymphatic vessels are absent in the hyaline cartilage [2]. Nutritional substances diffuse mostly through the synovial liquid of the joint cavity and provide tropism of the tissue [2]. The absence of innervation and blood supply are likely standing behind the poor healing capacity of articular cartilage [3] and without therapy focal articular cartilage defects lead to the degenerative changes of joint tissues and its functional loss [4,5]. 

New tissue engineering approaches are evolving to reconstruct articular cartilage damaged by a trauma or a disease [6]. Current surgical strategies for cartilage recovery are highly invasive and include a long period of subsequent rehabilitation of the patient. The list of clinical practices includes arthroscopic debridement [7], bone marrow stimulation techniques (such as subchondral bone perforation and microfracturing) [8], ACI (Autologous Chondrocyte Implantation) and its iterations [9], and mosaic osteochondral transplantation [10]. In addition, the methods of articular cartilage degeneration treatment are developed in the field of cell biology, genetic engineering, drug delivery systems, and growth factors to restore the damaged cartilage. However, the proposed approaches still have several limitations due to their high cost and complexity of implementation in clinical practice [11]. For this reason, the development of effective and optimal approaches to the biomaterial-based regeneration of hyaline cartilage is extremely urgent. 

Biomaterials, capable of replacing hyaline cartilage, are currently under development [12,13]. Biomaterials must carry specific properties and special matrix organization to withstand various mechanical stresses that occur during joint movements. At the same time, these materials must have a low friction coefficient, high abrasion resistance, and elasticity [2]. Biomaterials can be conventionally classified as synthetic and naturally occurring. Synthetic materials such as polycaprolactone, poly-L-lactic, and polyglycolic acids have numerous advantages including reproducibility, availability, low risk of biological contamination, etc. But their disadvantages, for example, acid decomposition products formation, low cellular interactions, and inappropriate intercellular signaling are still limiting the synthetic materials’ widespread application to date. On the other side, natural biopolymer-based scaffolds have several recognized advantages such as biocompatibility, structural similarity, and biological activity, absence of toxic side-products of biodegradation [14]. For example, the commercial collagen membrane Chondro-Gide^®^(Geistlich Pharma AG, Wolhusen, Switzerland) synthesized from the porcine tissue collagen fibers type I and III, is widely used in clinics to stimulate the natural healing of articular cartilage [15]. Producers recommend using the membrane for covering the «superclot» formed from the bone marrow cells, that enter the injured area through the bottom of the defect due to perforation of the subchondral bone before membrane implantation [16]. 

Our previous work has shown the absence of chondrogenesis while using Chondro-Gide^®^ (Geistlich Pharma AG, Wolhusen, Switzerland) membrane for covering a full-thickness cartilage defect in rats [17]. Here, we have compared the chondrogenic properties of several scaffolds, as well as characterized the response of the tissues surrounding the implant. 

## 2. Results

The defect area and the scaffolds were visible on macroscopic images both 2 and 4 months after full-thickness defects were performed and the scaffolds implanted (Figure 1). Two months after the surgery (Figure 1a–f), the congruence of the joint surface was recovered only in a group with implanted decellularized cartilage (Figure 1c) whereas 4 months after the surgery the congruence was restored in all groups except the one with cellulose-based scaffold (Figure 1l). Of note, for macro-images, only one animal per group was checked (knee dissembled) whereas all other animals were proceeded for microscopic analysis with sagittal sections through non-dissembled knees.

Hematoxylin and eosin (H&E (Merck KGaA, Darmstadt, Germany)) staining revealed that 2 months after surgery, in the absence of a scaffold (control) the damaged area was covered by newly formed tissue. This tissue differs from intact hyaline cartilage in the organization and density of cells, as well as in the amount of extracellular matrix (Figure 2(Aa)). The observed cells are small, polygonal in shape, and morphologically resemble mesenchymal cells or fibroblasts (Figure 2(Aa)). Implantation of membranes or decellularized cartilage led to partial filling of the defect zone with cells similar to those observed in the control (Figure 2(Ab–e)). The only exception was the cellulose scaffold, which was found to be migrating inside the subchondral bone, whereas the healing of the articular cartilage resembled control (Figure 2(Af)). Four months after implantation, H&E Merck KGaA, Darmstadt, Germany) revealed a large number of mesenchymal-like cells in all groups (Figure 2B). The non-resorbable cellulose-based scaffold gradually moved further from the damaged area into the underlying secondary ossification center (Figure 2 (Bf)) and its participation in the recovery processes looks unlikely.

The appearance of mesenchymal-like cells in all the groups (Figure 2A,B) implies their migration from other locations. The most likely source is the underlying bone marrow, known to contribute to cartilage regeneration during microfracture surgical procedures [18]. To test if the migrated cells are chondrogenic precursors, the expression of Sox9 protein was assessed. Immunostaining of the tissue sections did not reveal the SOX9-positive cells within the injured area neither 2 (Figure 3A) nor 4 months (Figure 3B) after surgery. In contrast, SOX-9 positive cells were observed in the intact cartilage tissue (Figure 3(Aa,Ba)). 

Since Sox9 is the marker for both chondroprogenitors and chondrocytes, the above results suggest that no hyaline cartilage is formed in the zone of the defect in either group despite the appearance of numerous cells. To check the formation of hyaline cartilage, the sections were stained with Safranin O/Fast Green (Merck KGaA, Darmstadt, Germany). Safranin O positive cartilage was not detected in the area of the defect in either control or scaffold-implanted groups (Figure 4). Notably, the presence of proteoglycans was not observed even in the group of implanted decellularized cartilage tissue (Figure 4(Ac,Bc)). It may indicate the necessity for constant deposition of proteoglycans by chondrocytes, or the loss of proteoglycans during decellularization. 

To determine the composition and organization of collagen fibers in the tissue formed in the injured area, the sections were stained with picrosirius red and assessed in polarized light. In intact cartilage, a superficial layer and subchondral bone stained red-orange, likely reflecting dense type I collagen fibers, whereas the main part of the hyaline cartilage comes greenish (Figure 5(Aa,Ba)). In all samples with implanted scaffolds, orange-red collagen fibers of the new tissue lie very tightly along the subchondral bone (Figure 5(Ab–Ag,Bb–Bg)). It indicates a strong birefringence, usually caused by the presence of collagen type I fibers. Therefore, this analysis revealed the presence of a large amount of type I collagen in the injured area. The analysis also showed the collagen fibers’ organization significantly different from the intact cartilage tissue.

## 3. Discussion

Degenerative diseases of the articular cartilage are widespread in the population and are one of the leading factors of population disablement [19]. The subchondral bone perforation is the most common clinical practice for small injuries. Such perforation (microfracture) is suitable for small defects (<2.5 cm^2^), as it leads to the formation of fibrocartilaginous tissue, which is relatively fragile and contains a large amount of type I collagen [20]. This approach is often combined with covering the injured area with a collagen membrane or implanting a scaffold into the damaged area in cases of a larger defect [18]. Here, we have demonstrated that this approach is sufficient to restore cartilage at the macro-level but does not lead to restoration of hyaline cartilage tissue or chondrocytes. None of the experimental groups formed full-fledged hyaline cartilage.

There are two parallel and interconnected strategies for cartilage repair, which are currently actively developing: (i) implantation of cell-free scaffolds and (ii) implantation of combined cells+scaffolds materials. Each has its own proves and cons. Cell-free methods imply the creating of material, that stimulates endogenous regenerative potential of cartilage or surrounding tissues. The absence of cell elements allows avoiding the high cost of the method, additional invasiveness, the duration of preparatory work, ethical and legal aspects. Such an approach is quite successfully applied in bone tissue regeneration [21]. On the other hand, articular cartilage is not innervated, vascularized, and has very limited regenerative potential, unlike bone tissue. This may indicate an insufficient number of endogenous stem cells [22]. Thus, cell-free approaches may be unsuccessful in cartilage tissue regeneration. Indeed, in recent research, Kumai T. et al. [23] used scaffolds that are compositionally close to native tissue. They are based on aggrecan, type II collagen, and hyaluronic acid. At the same time, the formation of the hyaline cartilage did not occur. This is comparable with the results presented here. In the other research, Dhollander et al. [24] used osteochondral plugs, which fill the defect without fissures and erosions formation but authors noted the formation of a disorganized extracellular matrix with a large number of fibroblasts in the newly formed tissue in biopsies after revision arthroscopy. 

We observed that cells of unknown origin are migrating into the injured site independent of the scaffold used. These cells are flat, oblong in shape cells and morphologically resemble fibroblasts or mesenchymal-like cells. Only occasionally cells with a rounded shape, morphologically resembling chondrocytes, were observed at the injury sites over time. At the same time, no accumulation of proteoglycans was observed in the formed tissue (as depicted by Safranin O) and these cells were not chondroprogenitors (because negative for SOX9 marker). The origin and the source of the cells filling the injury are currently unclear. There are several potential sources of cells, which can contribute to filling the injury site including bone marrow stromal cells from the underlying bone tissue, synovial cells, and superficial chondrocytes [18,25,26]. In our experiments, the source of the cells cannot be identified, but data clearly indicate that there are endogenous cells capable to migrate and fill up the cartilage defect in our model, but not capable to differentiate into chondrocytes independently of the various extracellular matrixes tested. These results suggest that scaffold implantation by itself is an insufficient method for the full-fledged recovery of hyaline cartilage structure. At the same time, the presence of invaded cells gives hope that a scaffold with growth factors being chondrogenic for these cells may work well. Along these lines, Y. Zhao [27] and Z. Jia [28] have recently shown that the combination of a scaffold consisting of cartilage extracellular matrix and microspheres of polylactide-glycolic acid with TGF-ß3 (transforming growth factor-beta 3) stimulated the new cartilage tissue formation in the almost entire injury site. The newly formed tissue contained round chondrocytes and showed higher levels of collagen type 2 and aggrecans [27,28]. Thus, cell-free scaffolds optimization by adding cytokines and/or growth factors is a promising strategy for the future. Currently, except for surgical approaches to cartilage repair, there is only one approved by the US Food and Drug Administration (FDA) and it is based on the combination of chondrocytes and matrix, the MACI method (matrix-induced autologous cartilage implantation) [29]. 

To summarize, the lack of hyaline cartilage recovery at the injured site, even after the implantation of decellularized articular cartilage, emphasizes a necessity to combine scaffolds with chondrogenesis inducers and/or cell therapy. At the same time, the observed appearance of cells morphologically looked like fibroblasts or mesenchymal cells in the injury site suggests that such a strategy may work. Characterization of the source and nature of these cells will make it possible to determine the cytokines and growth factors required for these cells’ differentiation.

## 4. Materials and Methods

### 4.1. Animals

Experiments on animals were approved by the Local Ethics Committee of the I.M. Sechenov First Moscow State Medical University (Sechenov University) (No. 07-17 of 13 September 2017) and were performed according to the requirements of the Guidelines for the maintenance and care of laboratory animals (Rules for the maintenance and care of laboratory rodents and rabbits, GOST 33216-2014). Adult (300–350 g) male Wistar rats were purchased from Pushino breeding house (Pushino, Russian Federation) and were maintained in a room with natural light and free access to fresh water and rodent chow.

### 4.2. Scaffolds

The commercial Chondro-Gide^®^ (Geistlich Pharma AG, Wolhusen, Switzerland) membrane (consists of type I and type III porcine collagens) was purchased from the manufacturer via our orthopedic clinic. Cellulose-based membrane, collagen type I and II membranes, as well as decellularized cartilage, were manufactured at the Department of Modern Biomaterials of the Institute of Regenerative Medicine of the I.M. Sechenov First Moscow State Medical University (Sechenov University) according to the protocols below.

### 4.3. Manufacturing of Type I Collagen Membrane (Membrane I)

For this membrane, type I collagen was obtained from the cattle dermis. The rinsed pieces of dermis were sequentially treated with an alkaline-salt solution of sodium hydroxide (NaOH (PanReac AppliChem, Darmstadt, Germany)) in a saturated solution of sodium sulfate (Na_2_SO_4_ (PanReac AppliChem, Darmstadt, Germany)), water, a solution of boric acid (H_3_BO_4_ (PanReac AppliChem, Darmstadt, Germany)), again with water. Then these pieces were dissolved in a 3% acetic acid solution (C_2_H_4_O_2_ (PanReac AppliChem, Darmstadt, Germany)) during amplified agitating. Thin dense plates, structured in formaldehyde (PanReac AppliChem, Darmstadt, Germany) vapor (CH_2_O), were obtained by lyophilization from the resulting solution. 

### 4.4. Manufacturing of Type II Collagen Membrane (Membrane II)

For this membrane, collagen II fibers were obtained from the cattle trachea. Prepared tracheal pieces were rinsed in a solution of sodium dihydrogen phosphate (NaH_2_PO_4_ (PanReac AppliChem, Darmstadt, Germany)), placed for 48 h in an alkaline-salt solution of sodium hydroxide (NaOH (PanReac AppliChem, Darmstadt, Germany)) in a saturated solution of sodium sulfate (Na_2_SO_4_ (PanReac AppliChem, Darmstadt, Germany)). Thereafter, the tracheal fragments were washed, dissolved, and dried in the same way as the dermis fragments.

### 4.5. Cartilage Decellularization

Hyaline cartilage for subsequent decellularization was obtained from the C-shaped rings of the cattle trachea. Obtained tracheal fragments were sequentially treated with a solution of disodium hydrogen phosphate (Na_2_HPO_4_ (PanReac AppliChem, Darmstadt, Germany)), purified water, aqueous-alcoholic sodium hydroxide solution (NaOH), and boric acid solution (H_3_BO_4_ (PanReac AppliChem, Darmstadt, Germany)). Thereafter the fragments were washed in purified water and dried by lyophilization. Finally, the decellularized cartilage sustained sterilization by gamma radiation. 

### 4.6. Cellulose Scaffold

Cellulose membrane was synthesized by a bacterial cellulose producer, the Gluconacetobacter hansenii GH-1/2008 (VKPM B-10547) strain [30]. The culture of bacteria strain GH-1/2008 was grown at 28 ± 2 °C for 5–7 days in the following medium: 30 g sucrose, 2.7 g disodium hydrogen phosphate (Na_2_HPO_4_ (PanReac AppliChem, Darmstadt, Germany)), 2 g dipotassium hydrogen phosphate (K_2_HPO_4_ (PanReac AppliChem, Darmstadt, Germany)), 3 g ammonium sulfate ((NH_4_)_2_SO_4_ (PanReac AppliChem, Darmstadt, Germany)), 1.15 g citric acid (C_6_H_8_O_7_ (PanReac AppliChem, Darmstadt, Germany)), 1% ethyl alcohol and 5 g yeast extract per 1 L of purified water. The derived 500 μm thick membranes were washed in running water, then in a 0.5 N solution of sodium hydroxide (NaOH (PanReac AppliChem, Darmstadt, Germany)) for 20 min at +40 °C, and neutralized with a 0.5 N acetic acid solution (C_2_H_4_O_2_ (PanReac AppliChem, Darmstadt, Germany)). The membranes were repeatedly washed with distilled water. The membranes were stored in distilled water until usage. 

### 4.7. Experimental Model: Knee Joint Articular Cartilage Full-Thickness Defect

We chose rats for a model of full-thickness osteochondral defect, because of their weak ability for spontaneous regeneration of articular cartilage as in humans. Animals were randomly allocated to experimental groups, with 5 animals in each group. In total, 30 rats were included in the experiment. The full-thickness defect was made in the femur epiphysis fossa of the knee joint [17]. The admission was implemented by medial parapatellar incision and abduction of patella aside. A defect was formed in the interstitial fossa with a hand cutter with a diameter of 2.0 mm and a depth until small blood secretions appeared at the bottom of the defect (Figure 6). In the control group, the surgery was performed the same way, but without scaffold implantation. Scaffolds were implanted in the knee joints in the following combinations: (a) articular cartilage defect without scaffold or membrane cover (control) on one knee and membrane I implantation to the other; (b) implantation of the Chondro-Gide^®^ (Geistlich Pharma AG, Wolhusen, Switzerland) membrane and membrane II; (c) implantation of the cellulose scaffold and decellularized cartilage. The animals were analyzed 2 and 4 months after surgery. Of note, 1 animal from each group was taken for visual macro-analysis and the knee was dissociated for femur and tibia whereas 4 others proceeded for sectioning with an intact knee.

### 4.8. Tissue Preparation, Histological Staining, and Microscopy 

The samples were fixed in 10% formalin, decalcified in EDTA (PanReac AppliChem, Darmstadt, Germany), frozen in O.C.T. (Sakura Finetek Japan Co.,Ltd., Tokyo, Japan), and then sectioned at 30 µm. The sections were stained by Hematoxylin and Eosin, Safranin O, and Fast Green (all Merck KGaA, Darmstadt, Germany). To assess the collagen fibers organization in newly formed tissues we used the staining with 0.1% solution of Direct Red 80 (365548-5G Merck KGaA, Darmstadt, Germany) in 1.3% picric acid aqueous solution. The sections were analyzed with the polarizing microscope [31]. Collagen I fibers form tightly packed bundles with strong birefringence and appear orange in polarized light. Collagen III fibers form thin bundles with weak birefringence and appear green in polarized light [31]. 

### 4.9. Immunostaining

Immunostaining was performed as described here [22]. SOX9 primary antibodies (HPA001758, Merck KGaA, Darmstadt, Germany) were used in dilution 1:100 and the corresponding secondary antibodies in dilution 1:400 (711-165-152 Jackson ImmunoResearch Europe Ltd, Cambridge House, St. Thomas’ Place, UK ). Nuclei were stained with DAPI (D8417, Merck KGaA, Darmstadt, Germany) for 20 minutes in concentration 10 µg per ml. The samples were examined by confocal scanning laser microscopy. Images were processed with microscope software (ZEN Black 2.0, Zeiss AG, Oberkochen, Germany).

## Figures and Tables

**Figure 1 ijms-23-00292-f001:**
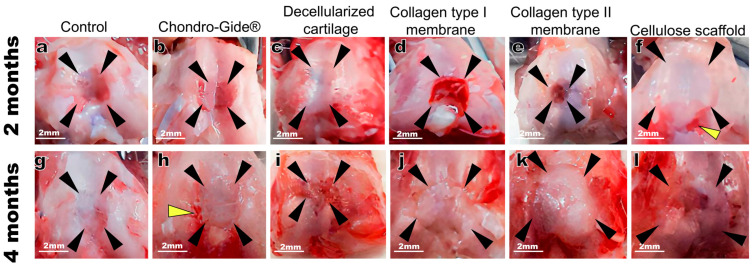
Macroscopic images of the knee joint with an osteochondral defect. The observation time is 2 months (**a**–**f**) and 4 months (**g**–**l**). The black arrows indicate the boundaries of the defect, the yellow arrows indicate the places of ulceration in the area of surgical intervention.

**Figure 2 ijms-23-00292-f002:**
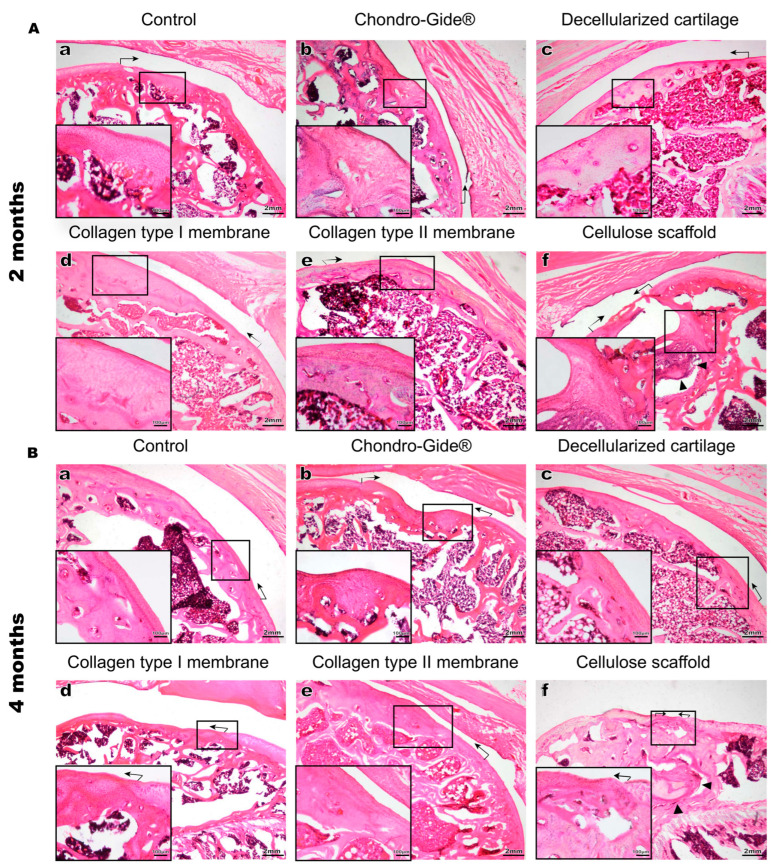
Histological preparations of the rat knee joint, H&E staining. The observation time is 2 months (**A**) and 4 months (**B**). **a**—control (no scaffold), **b**—Chondro-guide, **c**—decellularized cartilage, **d**—membrane made of collagen type I, **e**—membrane made of collagen type II, **f**—cellulose. The black arrows indicate the boundaries of the defect, the black arrowheads in (**Af**,**Bf**) indicate the part of the non-resorbed implant migrated into the secondary ossification center. Magnification of the image—×25, magnification of the enlarged area—×100.

**Figure 3 ijms-23-00292-f003:**
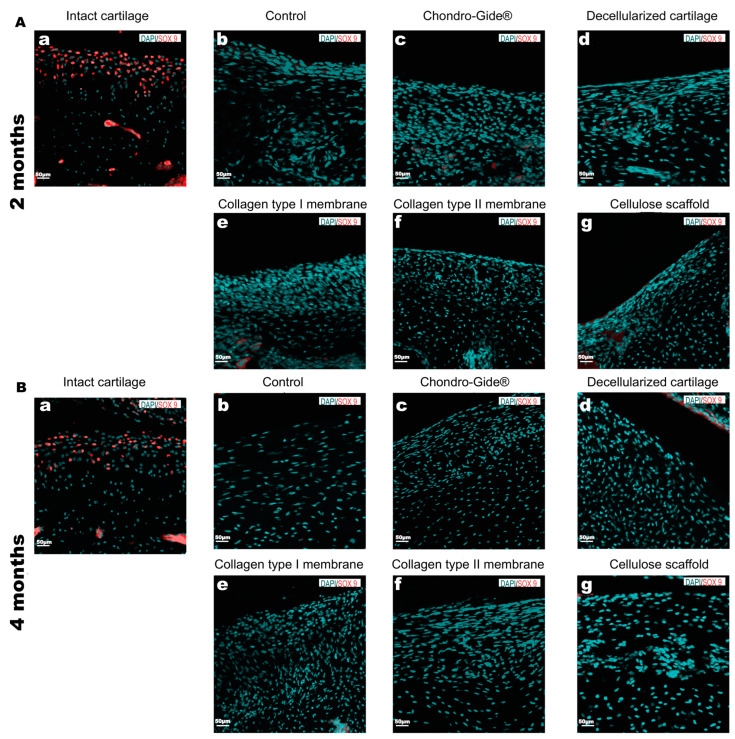
Immunofluorescence staining with antibodies to the transcription factor SOX9. The observation time is 2 months (**A**) and 4 months (**B**), magnification ×200. Nuclear localization of the signal is specific whereas cytoplasmic (e.g., (**Ac**,**Ae**)) is not specific and appeared due to high laser power used during confocal scans. Small labels **a**–**g** correspond to the text above the corresponding pictures.

**Figure 4 ijms-23-00292-f004:**
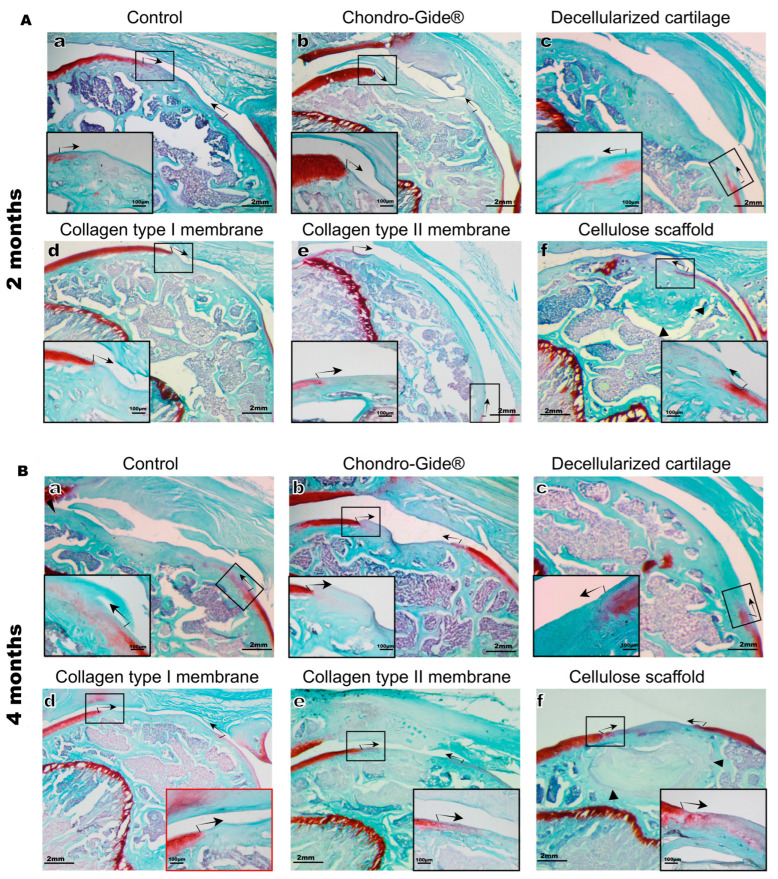
Histological preparations of the rat knee joint, staining with Safranin O/Fast Green. The observation time is 2 months (**A**) and 4 months (**B**). **a**—control (no scaffold), **b**—Chondro-guide, **c**—decellularized cartilage, **d**—membrane made of collagen type I, **e**—membrane made of collagen type II, **f**—cellulose. The black arrows indicate the boundaries of the defect, the black arrowheads indicate the part of the preserved implant. Magnification of the image—×25, magnification of the enlarged area—×100.

**Figure 5 ijms-23-00292-f005:**
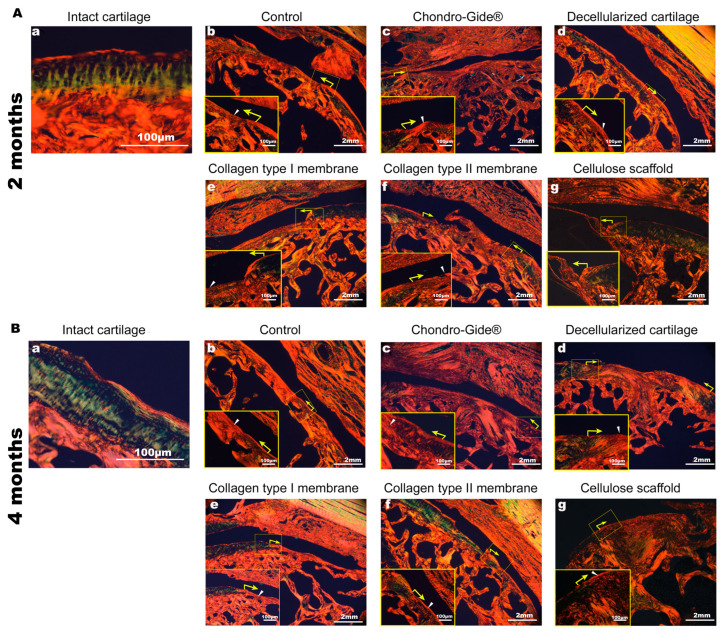
Organization of collagen fibers in the area of the transition of cartilage to the damage zone. Picrosirius red staining. The observation time is 2 months (**A**) and 4 months (**B**). Small labels **a**–**g** correspond to the text above the corresponding pictures. The yellow arrows indicate the border of the transition of the intact zone to the area of surgical intervention. The white arrowheads indicate the location of the collagen fibers in the defect area. Magnification of the image—×25, magnification of the enlarged area—×100.

**Figure 6 ijms-23-00292-f006:**
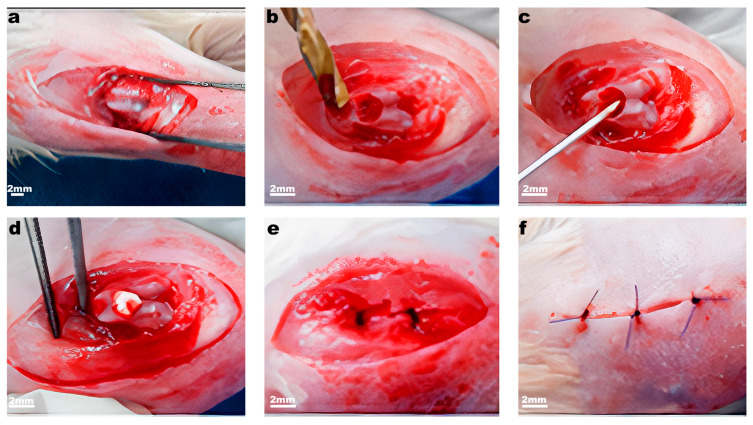
Stages of the operation for the formation of an osteochondral defect. (**a**) Arthrotomy; (**b**) creation of a full-thickness defect of articular cartilage in the interstitial fossa; (**c**) perforation of the subchondral layer (spongy substance); (**d**) implantation of the scaffold in the defect; (**e**,**f**) suturing of the operating wound.

## Data Availability

Not applicable.

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
