# Peer review of "Implantation of Various Cell-Free Matrixes Does Not Contribute to the Restoration of Hyaline Cartilage within Full-Thickness Focal Defects"

_ijms, 2021, doi:10.3390/ijms23010292_

Round 1

Reviewer 1 Report

Dear Authors,

I find your paper of vital importance in presenting the outlook on regeneration of articular cartilage utilizing both commercial and self-made decellularized scaffolds of different categories. 

The macroscopic and histological evaluation of the implantation zones is presented thoroughly with all atrifacts explained.

Study is well-concluded and clearly proves the initial thesis of lack of contribution of the implants to the regeneration of articular cartialge with full focal defects.

Considering above, I reccomend your article for publication.

Best regards, 

Reviewer

Author Response

We very much appreciate so high appraisal of our work by the reviewer. Thank you very much for the positive evaluation.

Reviewer 2 Report

Dear authors. Please find enclosed my comments.

In Materials and methods section please indicate the number of rats used.

Indicate the size of the defect and describe precisely how the defect is realized.

In the immunostaining section, indicate the secondary antibody used to reveal SOX9.

In the Results section, you need to modify and better describe the figures.

Figure 1 : Most photos are blurry. A scale bar is missing to have the size of the defect. It would be necessary to present the photos by 3 and to enlarge them. Use letters instead of numbers to standardize with the other figures.

Figure 2 : Most photos are blurry, we can't see well because of the reflections. For example photo l is too dark. A scale bar is missing to have the size of the defect.

Figure 3 : A scale bar is missing. The letters are at the bottom right while they are at the top left in figure 2 please standardize. In this figure there are black arrows and arrowheads but no yellow arrows please correct. What do the arrowheads represent?

Figure 4 : Put DAPI / SOX 9 at the top right and add letters (a, b ...... g) for A and B. We notice a discreet staining at 2 months for Chondro-GideÒ, decellularized cartilage and collagen type I membrane. Is it background noise? A scale bar is missing.

Figure 5 : A scale bar is missing. Put the letters (a, b ...... f) at the top left.

Figure 6 :  A scale bar is missing. Change white arrows to white arrowheads.

Best regards

Author Response

Dear authors. Please find enclosed my comments.

Dear reviewer, we appreciate your detailed examination of our article. We have modified the manuscript according to your suggestions. We hope that we answered all the questions sufficiently and we also apologize for uncertainties and lack of some details, which are now fixed. Please see below our point-to-point response.

In Materials and methods section please indicate the number of rats used.

The number of rats is now indicated (page 5, line 6)

Indicate the size of the defect and describe precisely how the defect is realized.

The size of the defect and the description of the defect formation is now indicated (page 5, lines 9-10)

In the immunostaining section, indicate the secondary antibody used to reveal SOX9.

The secondary antibody is now indicated (page 5, lines 29-30)

In the Results section, you need to modify and better describe the figures.

Figure 1 : Most photos are blurry. A scale bar is missing to have the size of the defect. It would be necessary to present the photos by 3 and to enlarge them. Use letters instead of numbers to standardize with the other figures.

We apologize for poor quality of the images. The image quality is now improved, enlarged and images arranged in the rows of 3. Also we added missing scale bars and changed numbers to letters according to your advice.

Figure 2 : Most photos are blurry, we can't see well because of the reflections. For example photo l is too dark. A scale bar is missing to have the size of the defect.

We  did the best we can to improve the image quality, and hope you find it acceptable now. Scale bars are now added.

Figure 3 : A scale bar is missing. The letters are at the bottom right while they are at the top left in figure 2 please standardize. In this figure there are black arrows and arrowheads but no yellow arrows please correct. What do the arrowheads represent?

We have added scale bars and changed the location of letters. In the figure description we changed the yellow arrows to black arrowheads. Black arrowheads indicate the part of the unresorbed scaffold. This is now clarified in page 9, lines 25-27.

Figure 4 : Put DAPI / SOX 9 at the top right and add letters (a, b ...... g) for A and B. We notice a discreet staining at 2 months for Chondro-GideÒ, decellularized cartilage and collagen type I membrane. Is it background noise? A scale bar is missing.

As suggested, DAPI/Sox9 indications are now relocated to the top right corner, small letters indicating each panel are now added, scale bars are now added. The observed discreet cytoplasmic staining at some panels, it is non-specific signal related to the high intensity of the laser used for these imaging to be sure not to miss any positive signal. Unspecificity is confirmed by cytoplasmic location of the signal, whereas SOX9 is characterised by nuclear localization of the signal (see positive control). This is now clarified at page 9, lines 31-32.

Figure 5 : A scale bar is missing. Put the letters (a, b ...... f) at the top left.

The figure is improved as suggested. Thank you. 

Figure 6 :  A scale bar is missing. Change white arrows to white arrowheads.

 Scale bars are added. ”White arrows is now changed” to ”white arrowheads” in the figure legend (page 10, line 9). We apologize for the inaccuracy.

We hope that all the reviewer’s concerns are now properly addressed and again thank the reviewer for helping us to improve the manuscript.

Round 2

Reviewer 2 Report

Dera authors

Thank you for your corrections. The manuscript can be published in this form.

Best regards